# Accelerating Multiple Intent Detection and Slot Filling via Targeted Knowledge Distillation

**Xuxin Cheng, Zhihong Zhu, Wanshi Xu,**
**Yaowei Li, Hongxiang Li, Yuexian Zou**[*]
School of ECE, Peking University, China
{chengxx, zhihongzhu, xwanshi, ywl, lihongxiang}@stu.pku.edu.cn
zouyx@pku.edu.cn

## Abstract

Recent non-autoregressive Spoken Language Understanding (SLU) models attracts increasing attention owing to the high inference speed. However, most of them still (1) suffer from the multi-modality problem since the prior knowledge about the reference is relatively poor during inference; (2) fail to achieve a satisfactory inference speed limited by their complex frameworks. To tackle these problems, in this paper, we propose a **T**argeted **K**nowledge **D**istillation **F**ramework (TKDF), which applies knowledge distillation to improve the performance. Specifically, we first train an SLU model as a teacher model, which has higher accuracy while slower inference speed. Then we introduce an evaluator and utilize the curriculum learning strategy to select proper targets for the student model. Experiment results on two public multi-intent SLU datasets demonstrate that our method can realize a flexible trade-off between inference speed and accuracy, achieving comparable performance to the state-of-the-art models while speeding up by over 4.5 times.

## 1 Introduction

Spoken language understanding (SLU) plays a pivotal and indispensable role in task-oriented spoken dialog systems (Young et al., 2013). It aims to understand the queries of users, which includes two subtasks: intent detection and slot filling (Tur and De Mori, 2011). Specifically, intent detection task aims to predict the intent of the given utterance and slot filling task aims to extract the additional information or constraints expressed in the utterance.

In real-world scenarios, it is common for an utterance to contain multiple intentions. In response to this challenge, Xu and Sarikaya (2013) and Kim et al. (2017) starts addressing the multi-intent SLU task. Gangadharaiah and Narayanaswamy (2019) makes the first attempt to use a multi-task framework to jointly model multiple intent detection and

---

[*] Corresponding author.

| Tokens | | Possibility | Reference |
|---|---|---|---|
| city | **intent:** | atis_city, atis_airport | atis_flight, atis_ground_fare |
| | **slot:** | B-toloc.city_name | I-toloc.city_name |
| service | **intent:** | atis_meal, atis_aircraft | atis_flight, atis_ground_fare |
| | **slot:** | I-transport_type | O |
| angeles | **intent:** | atis_airline, atis_airport | atis_flight, atis_ground_fare |
| | **slot:** | I-airport_name | I-city_name |

Table 1: Three examples of the multi-modality problem in non-autoregressive multi-intent SLU. We use the ground-truth intent label of the utterance as the intent of each token in the utterance.

slot filling. Recently, several graph-based models have shown promising results in jointly handling multiple intent detection and slot filling. Qin et al. (2020) proposes AGIF, which utilizes graph attention networks (GAT) (Velickovic et al., 2018) to predict fine-grained multi-intents by integrating intent information into the autoregressive decoding process of slot filling. Qin et al. (2021b) proposes the first non-autoregressive SLU model, achieving both speedup and the improved accuracy in multi-intent situations. Xing and Tsang (2022a) proposes the Co-guiding Net, which uses a two-stage framework achieving the mutual guidance between slot and intent. And Xing and Tsang (2022b) proposes Rela-Net, which incorporates a heterogeneous label graph to represent the statistical dependencies and hierarchies in rich relations, along with the recurrent heterogeneous label matching network to capture and leverage beneficial label correlations in an end-to-end manner. Song et al. (2022) proposes to leverage the statistical co-occurrence frequency between intents and slots as prior knowledge effectively in order to enhance the joint multiple intent detection and slot filling through constructing an intent-slot co-occurrence graph based on the entire training corpus to globally discover the correlation between intents and slots. Although existing non-autoregressive multi-intent SLU models have made promising progress (Qin et al., 2021b; Xing and Tsang, 2022a,b; Cheng et al., 2023c), we find that

most of them still suffer from two problems:

(1) **Multi-modality Problem.** As other non-autoregressive methods (Gu et al., 2018; Ma et al., 2019; Bao et al., 2021), non-autoregressive SLU methods also suffer from the multi-modality problem. As shown in Table 1, there might be multiple possible correct slots of a token and this problem also appears in intent detection with the widespread use of the token-level intent detection decoder. Despite (GAT) (Velickovic et al., 2018) is adopted to model the interaction between intents and slots, existing models still have little prior knowledge about the reference during the inference progress, leading to some errors that do not generally occur in the autoregressive models, which limits the performance of the non-autoregressive models.

(2) **Poor trade-off between inference speed and accuracy.** Although Co-guiding Net (Xing and Tsang, 2022a) and Rela-Net (Xing and Tsang, 2022b) achieve new state-of-the-art performance, they are both limited by their complex model architectures and thus cannot infer as fast as the past non-autoregressive SLU models, such as GL-GIN (Qin et al., 2021b) and LR-Transformer (Cheng et al., 2021a). For non-autoregressive models, inference speed is a crucial evaluation metric. As a result, it is crucial to implement a flexible trade-off between inference speed and accuracy.

For the first problem, a very common solution is to utilize knowledge distillation to first training an autoregressive model in the original training corpus and then applying the greedy outputs of the teacher model as the targets to train the non-autoregressive student model (Hinton et al., 2015; Gu et al., 2018; Stahlberg, 2020; Gou et al., 2021). However, there is still a lack of research on applying knowledge distillation to address the multi-modality problem in SLU. A very intuitive idea is to directly employ knowledge distillation to non-autoregressive SLU models, where the student SLU model could learn from the teacher SLU model. Unfortunately, our preliminary experimental results demonstrate that this method fails to solve multi-modality problem effectively, whose details could be seen in Sec.6.3.

We argue that the reason for these divergent results is that SLU is a classification task, while neural machine translation is a generative task (Liu and Lane, 2016; Zhu et al., 2023a). Considering that it has been verified that directly applying knowledge distillation to the non-autoregressive machine translation models will lead to the significant decrease in

terms of prediction accuracy for the low-frequency words (Ding et al., 2021; Du et al., 2021). In SLU task, owing to the limited categories of intents and slots, the decrease in terms of prediction accuracy of low-frequency words has a more serious impact on the overall performance.

For the second problem, knowledge distillation also seems to be an effective solution. Knowledge distillation has been widely used to improve reasoning speed without overly sacrificing performance in various tasks (Heo et al., 2019; Liu et al., 2020; Jiao et al., 2020). As a result, we decide to advance along this technical path and apply knowledge distillation to choose the teacher model with the higher accuracy and select a student model with the higher inference speed. By this means, the distilled SLU model would obtain a higher inference speed while almost maintaining the accuracy.

In order to address these above two problems at the same time, we try to improve the past methods instead of directly applying traditional knowledge distillation, thereby mitigating the multi-modality problem and achieving the trade-off between inference speed and accuracy. Motivated by the recent success of advanced knowledge distillation in other tasks (Huang and Wang, 2017; Liu et al., 2023; Gou et al., 2023a,b), we propose a **T**argeted **K**nowledge **D**istillation **F**ramework (TKDF). Specifically, we first train a multi-intent SLU model as the teacher SLU model with higher accuracy while slower inference speed. We regard its output as the distilled data. Then, we train a new SLU model with a simpler network on the distilled data as the evaluator and use a curriculum learning strategy to obtain the selected data by replacing the original distilled data with original data dynamically during the training process. Finally we train an SLU model with the same architecture as the evaluator on the selected data as the student SLU model. We also propose a metric to score the outputs, and the evaluator model assesses each utterance in the original training set through scoring the output intents and output slots. The utterances with higher scores are chosen as the targets, which typically contain more slight modality changes compared to the distilled data but show better prediction quality. We believe the selected utterances could be more safely exposed to the student model during training, which is beneficial for the overall performance.

We conduct all the experiments on two public benchmark multi-intent SLU datasets, MixATIS

and MixSNIPS (Hemphill et al., 1990; Coucke et al., 2018; Qin et al., 2020) and over three SLU architectures, Co-guiding Net (Xing and Tsang, 2022a), Rela-Net (Xing and Tsang, 2022b) and SSRAN (Cheng et al., 2022). Experiment results demonstrate that our method can realize a flexible trade-off between inference speed and accuracy, and further analysis also verifies the advantages of our framework. In summary, the core contributions of this work could be concluded as follows:

- We propose a non-autoregressive multi-intent SLU framework TKDF, which applies the targeted knowledge distillation method.

- The proposed method can achieve comparable performance to the state-of-the-art SLU models while achieving faster inference speed.

- Further analysis shows that distilling only 4% of the original data could help the student SLU model surpass its counterpart trained on the original data by a large margin.

## 2 Related Work

### 2.1 Intent Detection and Slot Filling

As two primary subtasks of SLU, intent detection and slot filling have sparked the research interests of an increasing number of researchers (Surdeanu, 2013; Mesnil et al., 2014; Hakkani-Tür et al., 2016; Zhang and Wang, 2016; Zhang et al., 2017; E et al., 2019; Liu et al., 2019; Qin et al., 2019, 2021a; Xie et al., 2023; Huang et al., 2023; Cheng et al., 2023a,d; Zhu et al., 2023c). As the strong interdependence between these two tasks is verified, an increasing number of joint models are achieving excellent performance (Chen and Yu, 2019; Zhang et al., 2019a; Qin et al., 2020; Bhathiya and Thayasivam, 2020; Qin et al., 2021b; Hui et al., 2021; Xing and Tsang, 2022a; Abro et al., 2022; Xing and Tsang, 2022b; Weld et al., 2022; Cheng et al., 2023b,e; Wu et al., 2023).

Recently, as the multi-intent SLU problem gradually obtains more and more attention, some SLU models based on graph attention are proposed gradually. AGIF proposed by Qin et al. (2020) is built upon a graph attention model, enabling the direct connections between the slot nodes of each token and all predicted intent nodes. This deliberate establishment of the correlations between slots and intents improves the grasp of their relationship, facilitating a deeper understanding. GL-GIN (Qin et al., 2021b) constructs a non-autoregressive graph interaction network conducting the parallel decoding

for slot filling. Co-guiding Net (Xing and Tsang, 2022a) introduces a two-stage framework to facilitate targeted enhancements through mutual guidance between two tasks. Rela-Net (Xing and Tsang, 2022b) leverages the heterogeneous label graph to further enhance the performance, incorporating statistical dependencies derived from co-occurrence patterns and the hierarchies in slot labels, as well as capturing rich relations among the label nodes.

Compared to previous works, the advantage of our framework is that we mitigate the negative impact of the multi-modality problem and achieve the trade-off between inference speed and accuracy.

### 2.2 Knowledge Distillation

Knowledge distillation (Hinton et al., 2015) is regarded as an effective method to mitigate the multi-modality problem in non-autoregressive models. In neural machine translation, sequence-level knowledge distillation (Kim and Rush, 2016) is widely used to substitute the original translations with the output generated by the pretrained teacher model. Zhou et al. (2020a) further explores knowledge distillation and proposes a range of techniques to fine-tune the complexity of a dataset in order to align it with the capacity of the non-autoregressive neural machine translation model, which aims to optimize the performance through effectively matching the intricacy of the dataset with the capabilities of the model. Shao et al. (2022) proposes diverse distillation with reference selection, which generates the new dataset with multiple high-quality references for each source sentence and selects the most fitting reference to train the non-autoregressive models, underscoring the importance of incorporating the diverse references in knowledge distillation.

Knowledge distillation has also proven its effectiveness in SLU task. Chen et al. (2022) proposes a self-distillation approach SDJN for improving joint modeling, allowing the model to leverage an auxiliary loop and exploits the interrelated connection between intent and slot information in depth. In our work, we utilize knowledge distillation to mitigate the multi-modality problem.

### 2.3 Curriculum Learning

Curriculum learning (Bengio et al., 2009) has been applied in numerous tasks, showcasing its versatility and efficacy (Braun et al., 2017; Graves et al., 2017; Matiisen et al., 2019). In the field of natural language understanding, Xu et al. (2020) uses curriculum learning and defines several easy exam-

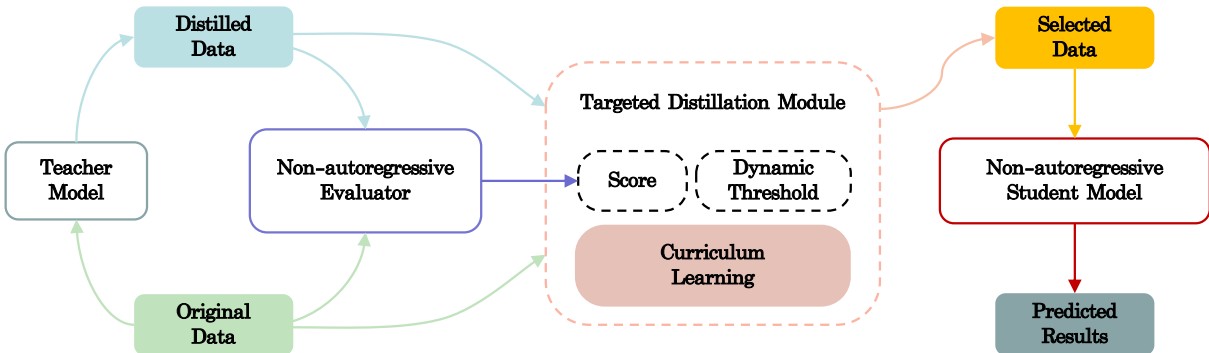

Figure 1: The architecture of our proposed framework, which consists of three components: (1) original data is fed into the teacher model to obtain the distilled data; (2) an evaluator constructs the selected data via curriculum learning according to the calculated score and the dynamic threshold; (3) an non-autoregressive student model is trained on the selected data and produces predicted results.

ples as those well solved by exact models. In the field of computer vision, Zhang et al. (2019b) proposes a novel collaborative self-paced curriculum learning regime, which leverages both the instance level prior-knowledge and the image level prior-knowledge in the unified, collaborative, and robust learning framework. Recently, Huang et al. (2020) embeds the idea of curriculum learning to the loss function to achieve the novel training strategy for deep face recognition, which mainly addresses easy samples during the early training stage and hard ones during the later stage.

## 3 Background

In general, intent detection and slot filling are two subtasks in SLU. Given an input utterance $x = (x_1, x_2, \ldots, x_n)$, where $n$ is the length of $x$, multiple intent detection is a multi-label classification task and the reference intent sequence is denoted as $\hat{y}^I = (\hat{y}^{(1,I)}, \hat{y}^{(2,I)}, \ldots, \hat{y}^{(m,I)})$, where $m$ is the number of intents in $x$. Slot filling is a sequence labeling task and the reference slot sequence is denoted as $\hat{y}^S = (\hat{y}^{(1,S)}, \hat{y}^{(2,S)}, \ldots, \hat{y}^{(n,S)})$. And we use $\hat{y}$ to denote the the union of $\hat{y}^I$ and $\hat{y}^S$.

Owing to the correlation between slots and intents, joint models are utilized to jointly optimize these two subtasks. The multi-intent detection objective and the slot filling objective are as follows:

$$\text{CE}(\hat{y}, y) = \hat{y} \log (y) + (1 - \hat{y}) \log (1 - y) \quad (1)$$

$$\mathcal{L}_I = -\sum_{i=1}^{n} \sum_{j=1}^{N_I} \text{CE}(\hat{y}_i^{(j,I)}, y_i^{(j,I)}) \quad (2)$$

$$\mathcal{L}_S = -\sum_{i=1}^{n} \sum_{j=1}^{N_S} \hat{y}_i^{(j,S)} \log \left( y_i^{(j,S)} \right) \quad (3)$$

where $N_I$ denotes the number of the single intent labels, $N_S$ denotes the number of slot labels, $\hat{y}_i^{(j,I)}$ denotes the reference intent, $y_i^{(j,I)}$ denotes its associated predicted intent, $\hat{y}_i^{(j,S)}$ denotes the reference slot, and $y_i^{(j,S)}$ denotes its associated predicted slot.

Following Qin et al. (2020, 2021b); Zhu et al. (2023b), the final joint objective is formulated as:

$$\mathcal{L} = \alpha \mathcal{L}_I + \beta \mathcal{L}_S \quad (4)$$

where $\alpha$ and $\beta$ are hyper-parameters.

## 4 Method

As shown in Figure 1, our TKDF could be divided into the following parts. First, we feed the original data into the teacher model with higher accuracy while slower inference speed to obtain the distilled data. Then, we utilize an evaluator module to construct the selected data, replacing the distilled data with the original data dynamically. Finally, we train the student model on the selected data.

The presence of multi-modality problem poses a challenge for the non-autoregressive SLU models, because it hinders accurate slot filling for certain tokens, consequently leading to misguided intent detection. Furthermore, the wide adoption of token-level intent detection decoders also exacerbates the negative impact of the multi-modality problem on intent detection performance.

In contrast, distilled data output by the teacher SLU model could simplify the training process of non-autoregressive models by reducing the target complexity. However, relying exclusively on distilled data is not the optimal approach, as it can be vulnerable to the errors made by the teacher model.

An intuitive approach is to leverage the strengths of both the original data and the distilled data by combining them. Motivated by Wang et al. (2021); Liu et al. (2023), we propose a non-autoregressive evaluator to determine whether the student model should be trained on the original data or the distilled data. The evaluator is first trained on the distilled data. Subsequently, we feed the original utterances into the evaluator, which assesses each prediction made on the original data. In cases where the non-autoregressive evaluator fails to produce outputs that closely align with the reference, we substitute the original labels with the corresponding distilled versions. Specifically, for the utterance $x$, we first get the output $y$ including the predicted intents and slots utilizing the non-autoregressive evaluator:

$$y = f_{\text{teacher}}(x) \tag{5}$$

Then we use the overall accuracy as the metric score to evaluate the original label $\hat{y}$. Predictions with the higher scores are considered more friendly for the the non-autoregressive student model. The score could be formulated as:

$$\text{score}(x, \hat{y}) = \text{Overall}(y, \hat{y}) \tag{6}$$

Inspired by recent success of curriculum learning (Graves et al., 2017; Doan et al., 2019; Zhou et al., 2020b; Luo et al., 2020), we apply curriculum learning to improve the performance. As the training progresses, the ratio of original data gradually diminishes. We establish a threshold $T$, wherein if the score surpasses this threshold, the distilled data will be replaced by the corresponding original data. The specific threshold $T_k$ is determined based on the current training step $k$:

$$T_k = \sigma\left(\frac{k}{K} - \frac{1}{2}\right) \tag{7}$$

where $\sigma(\cdot)$ denotes the sigmoid function and $K$ denotes the total number of updates.

## 5 Experiments

### 5.1 Datasets and Metrics

We evaluate our model using two public benchmark multi-intent SLU datasets[1], including the cleaned versions of MixATIS and MixSNIPS. MixATIS dataset is derived from the ATIS dataset (Hemphill et al., 1990), while MixSNIPS dataset is derived

---

[1] https://github.com/LooperXX/AGIF

from SNIPS dataset (Coucke et al., 2018). Compared to the single-domain MixATIS dataset, the MixSNIPS dataset is more complex due to its diverse intents and larger vocabulary. The statistics of datasets used are shown in Table 2.

| Dataset | MixATIS | MixSNIPS |
|---|---|---|
| Vocabulary Size | 722 | 11241 |
| Intent Categories | 17 | 6 |
| Slot Categories | 116 | 71 |
| Training Set Size | 13162 | 39776 |
| Validation Set Size | 756 | 2198 |
| Test Set Size | 828 | 2199 |

Table 2: Dataset statistics.

For all the experiments, we select the SLU model which works the best on the *dev* set and then evaluate it on the *test* set. For slot filling, we measure the performance using the F1 score, which provides an evaluation of the ability to correctly identify and fill slots in the input utterances. For intent detection, we utilize the accuracy score, which measures the ability to accurately predict the intent behind each utterance. Furthermore, we evaluate the overall accuracy of the utterance-level semantic frame parsing, which accounts for the correctness of all predicted metrics within an utterance.

### 5.2 Baselines

We compare our method with the following strong multi-intent SLU baselines:

- Attention BiRNN (Liu and Lane, 2016): an attention-based neural network model for joint intent detection and slot filling.
- Slot-Gated (Goo et al., 2018): a joint SLU model with a slot gate that focuses on learning the relationship between slot and intent to obtain better semantic frame result.
- Bi-Model (Wang et al., 2018): a Bi-model to consider the impact between slot and intent.
- SF-ID (E et al., 2019): a joint model to establish the connections for slot and intent.
- Stack-Propagation (Qin et al., 2019): a stack-propagation framework to incorporate intent detection to guide slot filling.
- AGIF (Qin et al., 2020): an LSTM-based adaptive framework to achieve the multi-intent information integration.
- LR-Transformer (Cheng et al., 2021a): a joint SLU model based on the Transformer with the

| Models | MixATIS | | | MixSNIPS | | | Speedup |
|---|---|---|---|---|---|---|---|
| | Overall(Acc) | Slot (F1) | Intent(Acc) | Overall(Acc) | Slot(F1) | Intent(Acc) | |
| Attention BiRNN (Liu and Lane, 2016) | 39.1 | 86.4 | 74.6 | 59.5 | 89.4 | 95.4 | 1.0× |
| Slot-Gated (Goo et al., 2018) | 35.5 | 87.7 | 63.9 | 55.4 | 87.9 | 94.6 | 0.9× |
| Bi-Model (Wang et al., 2018) | 34.4 | 83.9 | 70.3 | 63.4 | 90.7 | 95.6 | 1.1× |
| SF-ID (E et al., 2019) | 34.9 | 87.4 | 66.2 | 59.9 | 90.6 | 95.0 | 1.2× |
| Stack-Propagation (Qin et al., 2019) | 40.1 | 87.8 | 72.1 | 72.9 | 94.2 | 96.0 | 1.4× |
| AGIF (Qin et al., 2020) | 40.8 | 86.7 | 74.4 | 74.2 | 94.2 | 95.1 | 1.1× |
| LR-Transformer (Cheng et al., 2021b,a) | 43.3 | 88.0 | 76.1 | 74.9 | 94.4 | 96.6 | 10.8× |
| GL-GIN (Qin et al., 2021b) | 43.0 | 88.2 | 76.3 | 73.7 | 94.0 | 95.7 | 11.2× |
| SDJN (Chen et al., 2022) | 44.6 | 88.2 | 77.1 | 75.7 | 94.4 | 96.5 | 0.8× |
| Co-guiding Net (Xing and Tsang, 2022a) | 51.3 | 89.8 | 79.1 | 77.5 | 95.1 | 97.7 | 2.4× |
| GISCo (Song et al., 2022) | 48.2 | 88.5 | 75.0 | 75.9 | 95.0 | 95.5 | 0.8× |
| DARER$^2$ (Xing and Tsang, 2023) | 49.0 | 89.2 | 77.3 | 76.3 | 94.9 | 96.7 | 2.5× |
| ReLa-Net (Xing and Tsang, 2022b) | 52.2 | 90.1 | 78.5 | 76.1 | 94.7 | 97.6 | 2.8× |
| SSRAN (Cheng et al., 2022) | 48.9 | 89.4 | 77.9 | 77.5 | 95.8 | 98.4 | 4.2× |
| Co-guiding w/ Standard KD | 48.4[†] | 88.5[†] | 76.6[†] | 74.8[†] | 94.3[†] | 96.2[†] | 10.8× |
| Co-guiding w/ TKDF (ours) | **50.8**[†] | **89.6**[†] | **78.8**[†] | **77.3**[†] | **94.6**[†] | **97.4**[†] | 10.8× |
| Rela-Net w/ Standard KD | 48.8[†] | 88.9[†] | 76.3[†] | 74.4[†] | 94.1[†] | 95.9[†] | 14.3× |
| Rela-Net w/ TKDF (ours) | **51.2**[†] | **89.8**[†] | **78.4**[†] | **75.9**[†] | **94.2**[†] | **97.0**[†] | 14.3× |
| SSRAN w/ Standard KD | 46.8[†] | 88.2[†] | 76.8[†] | 74.5[†] | 94.8[†] | 96.6[†] | 19.3× |
| SSRAN w/ TKDF (ours) | **48.5**[†] | **89.2**[†] | **77.6**[†] | **77.2**[†] | **95.4**[†] | **98.1**[†] | 19.3× |

Table 3: Main results. † denotes our model significantly outperforms baselines with $p < 0.01$ under t-test. "w/ Standard KD" means that we use the same standard knowledge distillation method as Gu et al. (2018).

layered refined mechanism.

- GL-GIN (Qin et al., 2021b): a LSTM-based global-locally graph interaction framework.
- SDJN (Chen et al., 2022): a joint SLU model using self knowledge distillation.
- Co-guiding Net (Xing and Tsang, 2022a): a two-stage model achieving the mutual guidance between slot filling and intent detection.
- GISCo (Song et al., 2022): an SLU model constructing co-occurrence graph based on the entire corpus to make use of the co-occurrence frequency between slot and intent.
- DARER$^2$ (Xing and Tsang, 2023): an SLU model utilizing the relational Transformer to achieve fine-grained temporal modeling.
- ReLa-Net (Xing and Tsang, 2022b): an SLU model exploiting the label typologies and relations among the labels.
- SSRAN (Cheng et al., 2022): an SLU model with a scope recognizer and a result network.

### 5.3 Training Settings

We select three models as the teacher models, including Co-guiding (Xing and Tsang, 2022a), Rela-Net (Xing and Tsang, 2022b) and SSRAN (Cheng et al., 2022). And we select GL-GIN (Qin et al., 2021b) as the evaluator model. We use Adam opti-

mizer (Kingma and Ba, 2015) with $\beta_1 = 0.9, \beta_2 = 0.98$ to optimize parameters in our model. The learning rate warms up to $5e - 4$ and then decays with a inverse square-root schedule. The total number of updates $K$ is set to 4000, and the value of label smoothing is set to 0.1. For other hyperparameters, we follow the values provided in the corresponding paper. All the experiments are conducted on a single Nvidia V100 GPU.

### 5.4 Main Results

Table 3 demonstrates the experiment results, from which we find that TKDF could achieve the comparable performance to three strong baselines, including Co-guiding Net, Rela-Net and SSRAN while increasing the inference speed by a large margin. This is due to the fact that our TKDF mitigates the negative impact of multi-modality problem by introducing an evaluator to obtain the selected data by replacing the original distilled data with original data dynamically in the training process.

## 6 Model Analysis

### 6.1 Effect of Dynamic Threshold

To evaluate the effectiveness of the dynamic threshold strategy, we conduct an experiment comparing its performance with the use of a fixed threshold. Co-guiding Net is employed as the teacher model,

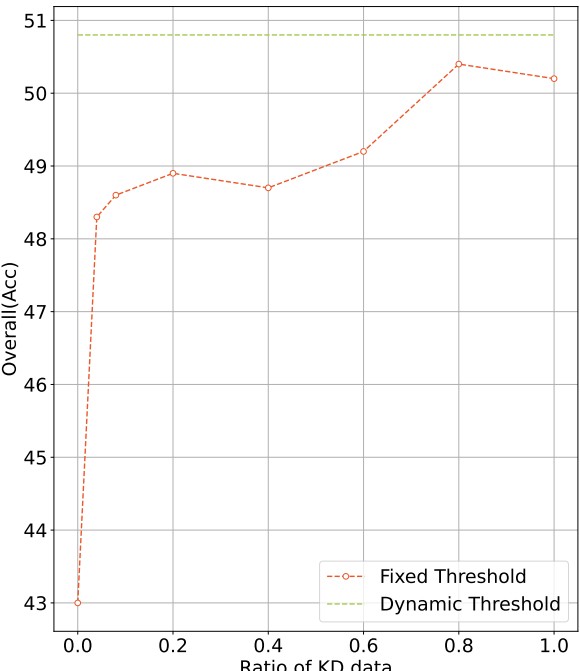

Figure 2: Overall accuracy of Co-guiding Net + TKDF on the MixATIS dataset with the fixed threshold and the proposed dynamical threshold.

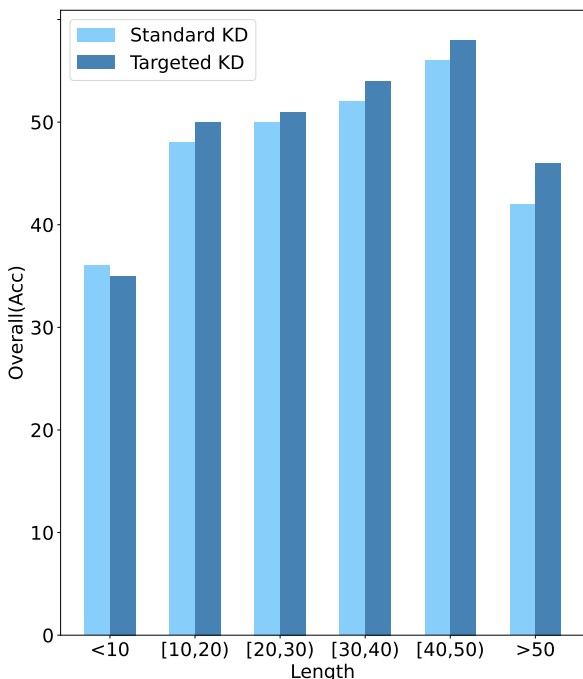

Figure 3: Overall accuracy of Co-guiding Net + TKDF examples of different lengths on the MixATIS dataset.

while GL-GIN serves as the student model. In the experiment, we set a fixed threshold value, denoted as $T$, to ensure that the training data remained constant throughout the training process.

As shown in Figure 2, the model demonstrates significant improvements and surpasses its counterpart by 14.6% in terms of overall accuracy, utilizing only 4% of the refined data. This result further validates the effectiveness of our targeted distillation method. By selectively filtering original data, the training data complexity could be reduced, leading to the enhanced performance. It is also worth noting that as the percentage of distilled data increases, the rate of performance growth begins to gradually diminish. The superiority of the dynamic threshold strategy over all fixed threshold settings not only shows the advantages of curriculum learning strategy but also enhances the flexibility of our proposed overall model architecture.

### 6.2 Effect of Targeted Knowledge Distillation

To verify the effectiveness of targeted distillation strategy, we compare it against the standard distillation. Our experiments are conducted on utterances of varying lengths, applying Co-guiding Net as the teacher model and GL-GIN as the student model. Figure 3 showcases the experimental results, specifically the overall scores for utterances of different

lengths. Notably, we observe that longer utterances benefit more from targeted knowledge distillation.

Intuitively, longer utterances are more prone to errors throughout the distillation process. Consequently, leveraging real data aids the student model in avoiding or rectifying these errors, thereby enhancing its ability to effectively model the longer utterances. We also observe a slight performance decrease for utterances with fewer than 10 tokens. It is owing to the fact that shorter utterances typically exhibit higher average scores, resulting in prolonged exposure to the non-autoregressive student model. However, the extended exposure to the original data may inadvertently introduce confounding factors during model training, as the model becomes influenced by the inherent multi-modality problem present in the original data.

### 6.3 Case Study

To further demonstrate the superiority of our model relative to the previous works on the multiple intent detection and slot-filling problems, we provide a case study as illustrated in Figure 4 which includes the prediction results of a utterance sequence under different distillation strategies. From the case, we can find that different distillation strategies can improve the original predicted incorrect utterances to various degrees. The prediction results for both slots and intents without distillation have some er-

**Models**

**Utterance:** tell me about the ground transportation from nashville airport

**Ref.**
Slot: O O O O O O O **B-airport_name** **I-airport_name**
Intent: atis_airfare atis_day_name atis_ground_service

**w/o KD**
Slot: O O O O O O O **B-fromloc.airport_name** **I-fromloc.airport_name**
Intent: atis_flight atis_ground_service atis_day_name

**Standard KD**
Slot: O O O O O O O **B-airport_name** **I-airport_name**
Intent: atis_airfare atis_ground_service atis_day_name

**TKDF**
Slot: O O O O O O O **B-airport_name** **I-airport_name**
Intent: atis_airfare atis_day_name atis_ground_service

Figure 4: A case study of our framework under different distillation strategies. Intents and slots in red are those that are predicted incorrectly, and intents in blue are those that are missed in prediction.

rors due to the multi-modality problem. The distilled data encompasses some valuable information derived from high-accuracy teachers, and employing a standard distillation strategy could partially alleviate aforementioned errors. However, the targeted distillation strategy yields highly favorable outcomes by accurately predicting intentions and slots. We attribute this success to its capability to not only amalgamate information from the teachers but also retain the original and crucial information.

## 7 Conclusion

In this paper, we propose TKDF, a simple yet effective method to solve the multi-modality problem of non-autoregressive SLU. We introduce an evaluator and apply a curriculum learning strategy to select proper targets for the student model. Experiments and analysis demonstrate the effectiveness of TKDF, which could achieve a flexible trade-off between inference speed and accuracy.

## Ethics Statement

We conduct all the experiments using two public datasets. These datasets do not contain any personally identifiable information, offensive content, or data that can be used to identify individual people.

## Limitations

Although our TKDF achieves a flexible trade-off between inference speed and accuracy, it does not change the inherent structure. We suppose that the understanding module in existing SLU models is not sufficient enough and limits the performance to some extent. In the future, we plan to explore more techniques to further improve the performance.

## Acknowledgements

We thank all anonymous reviewers for their constructive comments. This paper was partially supported by Shenzhen Science & Technology Research Program (No: GXWD20201231165807007-20200814115301001) and NSFC (No: 62176008).

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
