# OpenReview forum: "Accelerating Multiple Intent Detection and Slot Filling via Targeted Knowledge Distillation"
_EMNLP/2023/Conference — EMNLP 2023 Findings_

### Official Review · Reviewer_3Dd1 · 2023-08-04

**Soundness:** 2

**Excitement:**

3: Ambivalent: It has merits (e.g., it reports state-of-the-art results, the idea is nice), but there are key weaknesses (e.g., it describes incremental work), and it can significantly benefit from another round of revision. However, I won't object to accepting it if my co-reviewers champion it.

**Paper Topic And Main Contributions:**

The paper proposes a knowledge distillation method to alleviate the multi-modality and accuracy problem in non-autoregressive multi-intent SLU methods. They show comparable performance to the state-of-the-art models while using only few original data to train a student model.

**Questions For The Authors:**

1. Could you explain what is the multi-modality problem of multiple slots illustrated from Table 1? Is there just one slot possibility for each example?
2. Could you explain and define what original data, distilled data, and selected data are and how you use the benchmark datasets for each of these?
3. From L119-L121, do you mean you take the predicted labels of the teacher model and their inputs to train an evaluator? Like pseudo-labeling?
4. Is the distilled data totally separate from the original data or the same data but with the predicted outputs from the teacher model? If that is the case, is it coming from validation or test set to train the evaluator? Then what is the accuracy for the evaluator? What percentage of original data is left to train the final student model?

**Reasons To Accept:**

1. The paper proposes a curriculum learning method to select high-quality data for averting multi-modality problem and achieving more competitive performance.
2. Many quantitative and qualitative experiments are conducted to verify the performance of the model and achieve comparable performance to the state-of-the-art models.

**Reasons To Reject:**

1. It’s hard to follow the framework details where additional rewriting and explanation would be suggested. It took me several times to understand the key idea of several sentences like what constitutes distilled data or selected data.
2. The data usage for each stage of training and testing is vaguely explained which makes it hard to catch up the idea for data filtering process. And some terms are rarely defined and explained.

**Reproducibility:**

3: Could reproduce the results with some difficulty. The settings of parameters are underspecified or subjectively determined; the training/evaluation data are not widely available.

**Reviewer Confidence:**

3: Pretty sure, but there's a chance I missed something. Although I have a good feel for this area in general, I did not carefully check the paper's details, e.g., the math, experimental design, or novelty.

**Typos Grammar Style And Presentation Improvements:**

L60 design → desgins
L118 inference inference → inference
L154 Can an → can

---

> ### Author Rebuttal · Authors · 2023-08-29
>
> We believe that you may misunderstand the paper. We should emphasize that both the teacher model and the evaluator have been trained on the benchmark dataset. We hope you will clear up the misunderstandings based on our response and consider improving your score. For the typos, we will fix them.
>
> Q1: What is the multi-modality problem of multiple slots illustrated from Table 1?
>
> A1: Thanks for your question! Multi-modality problem is a common problem in non-autoregressive models. For a word, there may be many corresponding slots. Since non-autoregressive models do not depend on the previous output, the prediction may be other corresponding slots, which leads to errors. In Table 1, I-transport_type and O are both corresponding slots of service in different utterances. However, the model incorrectly predicts O as I-transport_type.
>
> Q2: What is original data, distilled data, and selected data?
>
> A2: As we mentioned in L119-125. The original data is the data from the original benchmark dataset, the distilled data is the output of the teacher model with higher accuracy while slower inference speed, and the selected data is obtained by Eq. 7 in L347. You could read the Algorithm 1 in Page 5 to understand it more clearly.
>
> Q3: Do you mean you take the predicted labels of the teacher model and their inputs to train an evaluator?
>
> A3: As we mentioned in L390-391. We use a trained GL-GIN on the benchmark dataset as the evaluator. We will further emphasize this point in the final version.
>
> Q4: Is the distilled data totally separate from the original data or the same data but with the predicted outputs from the teacher model? What percentage of original data is left to train the final student model?
>
> A4: The distilled data is the output of the teacher model. Note that both the teacher model and the evaluator have been trained on the benchmark dataset. All the original data is used to replace the distilled data dynamically during the training stage, according to Equation 7 in L347.

---

### Official Review · Reviewer_y4ok · 2023-08-05

**Soundness:** 3

**Excitement:**

3: Ambivalent: It has merits (e.g., it reports state-of-the-art results, the idea is nice), but there are key weaknesses (e.g., it describes incremental work), and it can significantly benefit from another round of revision. However, I won't object to accepting it if my co-reviewers champion it.

**Paper Topic And Main Contributions:**

The paper introduces a Targeted Knowledge Distillation Framework (TKDF) designed to enhance the performance of non-autoregressive Spoken Language Understanding (SLU) models. To achieve this, the authors first train a multi-intent autoregressive SLU model as the teacher model. Then, they proceed to train a non-autoregressive SLU model on the distilled data, which acts as an evaluator to determine when to replace the distilled data with the original while training another non-autoregressive student SLU model.

The authors observed that their approach resulted in a speedup over autoregressive SLU models while also narrowing the performance gap between autoregressive and non-autoregressive SLU models.

**Questions For The Authors:**

Check Reasons To Reject

**Reasons To Accept:**

1. Significant speedup compared to autoregressive Spoken Language Understanding models while achieving comparable performance.
2. Paper is mostly well written and does a good job to motivate their approach.

**Reasons To Reject:**

1. The performance improvement does not appear significant to me. I'm unsure which baselines are used for the p-tests in Table 3. According to the text in 4.4, the baselines should have been Co-guiding Net, 407 Rela-Net, and SSRAN. However, in some cases, such as Co-guiding Net vs. Co-guiding Net w/ TKDF on MixSNIPS, the proposed approach doesn't seem to outperform these baselines. Am I misunderstanding something here?

2. Section 3.1 was a bit hard to read, and I found myself confused about the difference between Slight/Serious Modality Change and Slight/Serious Mistake. The whole distinction seems very subjective to me.

3. One major weakness, in my opinion, is the lack of a strong connection between their motivation, i.e., Multi-modality, and the results. The discussion in section 5.3, with just one error example, is not very convincing.

4. The novelty of their approach seems rather incremental to me, especially considering I'm not entirely convinced if their approach truly addresses the multi-modality aspect. I feel more detailed ablation study is needed here.

**Reproducibility:**

3: Could reproduce the results with some difficulty. The settings of parameters are underspecified or subjectively determined; the training/evaluation data are not widely available.

**Reviewer Confidence:**

3: Pretty sure, but there's a chance I missed something. Although I have a good feel for this area in general, I did not carefully check the paper's details, e.g., the math, experimental design, or novelty.

---

> ### Author Rebuttal · Authors · 2023-08-29
>
> We appreciate your constructive feedback.
>
> Q1: The performance improvement does not appear significant to me.
>
> A1: As stated in the title of the article, the goal of this paper is to accelerate multiple intent detection and slot filling. Although the inference speed of non-autogressive models is usually faster than that of autogressive models, it's also common that non-autogressive models perform worse than autogressive models. We aim to improve the inference speed while causing as little performance degradation as possible. The models with our proposed TKDF outperforms the models with Standard KD and achieves a $\textbf{much faster}$ inference speed with the original models. Therefore, the performance improvement is significant.
>
> Q2: The whole distinction seems very subjective.
>
> A2: Thanks for your question. We provide 4 situations where the distilled data is different from the original ones to help readers understand the motivation more clearly. Actually, you can also make other distinctions, which will not affect the results.
>
> Q3: The discussion in section 5.3, with just one error example, is not very convincing.
>
> A3: Thanks for your advice. Due to the limited space, we only provide one error example. We will add more analysis in the final version.
>
> Q4:  More detailed ablation study is needed here.
>
> A4: Thanks for your advice.  We will add more detailed ablation studies in the final version.

---

### Official Review · Reviewer_bSiD · 2023-08-05

**Soundness:** 3

**Excitement:**

3: Ambivalent: It has merits (e.g., it reports state-of-the-art results, the idea is nice), but there are key weaknesses (e.g., it describes incremental work), and it can significantly benefit from another round of revision. However, I won't object to accepting it if my co-reviewers champion it.

**Paper Topic And Main Contributions:**

The paper proposes TKDF, a model for non-autoregressive multi-intent Spoken Language Understanding using targeted knowledge distillation. The proposed model accelerates reference speed while achieving results similar to the state-of-the-art models. Further analysis reveals that by only 4% of the original data can help the student model outperform baseline student model trained on the entire original data by a significant margin.

**Questions For The Authors:**

1. In the result, the authors claimed that TKDF "accelerates the inference speed by 4.5 times, 5.1 times and 4.6 times Co-guiding, Rela-Net and SSRAN, respectively. Since inherent structure of the model is not changed, does the speed improvement come from parallel computing?
2. In Section 5.2, does the MixSNIPS show similar phenomena that only 4% of the original data can help the student model outperform 14.6% overall accuracy and "80% distilled data slightly outperforms the model trained on fully distilled data" as on MixATIS. MixATIS is a simpler task which contains a lot of repeated expressions. MixSNIPS is a more challenging benchmark while MixATIS also need further analysis.
3. Why dynamic threshold in Figure 3 is a constant number?

**Reasons To Accept:**

1. the paper proposes a novel framework using targeted knowledge distillation and curriculum learning that achieves trade-offs of inference speed and accuracy.
2. Experimental results shows it outperforms standard knowledge distillation and improves inference speed by a large margin.

**Reasons To Reject:**

1. The motivation of multi-modality problem is not well addressed in empirical experiments. Metrics are needed for 4 situations in Table 2 to show its performance.
2. It requires more evidence to show the necessity of using teacher-student model. Analyses are missing on MixSNIPS, which is a more challenging benchmark and has better generality.

**Reproducibility:**

4: Could mostly reproduce the results, but there may be some variation because of sample variance or minor variations in their interpretation of the protocol or method.

**Reviewer Confidence:**

5: Positive that my evaluation is correct. I read the paper very carefully and I am very familiar with related work.

---

> ### Author Rebuttal · Authors · 2023-08-29
>
> We appreciate your constructive feedback.
>
> Q1: Metrics are needed for 4 situations in Table 2 to show its performance.
>
> A1: Thanks for your advice. Due to the limited space, we only report the results for the entire dataset. We will add the results for each of the 4 situations in Table 2 to further show the performance.
>
> Q2: Analyses are missing on MixSNIPS, which is a more challenging benchmark and has better generality.
>
> A2: Thanks for your advice. Due to the limited space, we follow previous work [1] to only conduct analyses on MixATIS. We will take your advice and add the analyses on MixSNIPS in the final version.
>
> Q3: Since inherent structure of the model is not changed, does the speed improvement come from parallel computing?
>
> A3: Thanks for your question. Yes, you are right. The speed improvement comes from parallel computing.
>
> Q4: Does the MixSNIPS show similar phenomena that only 4% of the original data can help the student model outperform 14.6% overall accuracy and "80% distilled data slightly outperforms the model trained on fully distilled data" as on MixATIS?
>
> A4: Thanks for your question. Only 3% of the original data can help the student model outperform 4.2% overall accuracy and 75% distilled data slightly outperforms the model trained on fully distilled data on MixSNIPS. Note that since the overall accuracy of GL-GIIN is 73.7 on MixSNIPS, which is relatively high, the improvement in overall accuracy on MixSNIPS is not as significant as with MixATIS. We will add these results on MixSNIPS in the final version.
>
> Q5: Why dynamic threshold in Figure 3 is a constant number?
>
> A5: Thanks for your question. The dynamic threshold in Figure 3 means that the threshold is calculated by using the Equation 8 on line 354, so the corresponding overall accuracy is a constant number.
>
> [1] Xing, Bowen, and Ivor Tsang. "Group is better than individual: Exploiting Label Topologies and Label Relations for Joint Multiple Intent Detection and Slot Filling." Proceedings of the 2022 Conference on Empirical Methods in Natural Language Processing. 2022.

---

### Meta-Review · Area_Chair_kYuK · 2023-09-18

**Recommendation:** 3

**Metareview:**

This paper addresses the multi-modality Spoken Language Understanding (SLU) problem. The authors aim to enhance the performance of a small-sized, fast-inference-speed student model (which is non-autoregressive) by distilling knowledge from a larger, slower-inference-speed teacher model (which is autoregressive). A novel "Targeted Knowledge Distillation Framework" (TKDF) is introduced. This framework incorporates an evaluator and employs a curriculum learning approach to choose appropriate targets for the student model. Experiments reveal that the distilled student model, though smaller in size, matches the performance of the larger model while being 4.5 times faster in inference.

**Soundness Scores**: (3, 3, 2).
The paper's soundness is good. It includes results from two benchmarks. In the rebuttal phase, the authors introduced another benchmark result. TKDF exhibits a performance advantage over standard knowledge distillation, even though their inference speeds are comparable. However, to bolster the paper's claims, the following enhancements are suggested:
1. A comprehensive analysis of the model across all test benchmarks.
2. Inclusion of detailed experimental outcomes in the appendix.

**Excitement Scores**: (3, 3, 3).
From an excitement standpoint, the paper doesn't particularly stand out. The rationale for employing TKDF to tackle the SLU issue isn't convincingly articulated in the manuscript. As a result, the proposed TKDF appears to be a modest incremental improvement over existing methods.

---

### Decision · Program_Chairs · 2023-10-07

**Decision:**

Accept-Findings

**Comment:**

This paper addresses the multi-modality Spoken Language Understanding (SLU) problem. The authors aim to enhance the performance of a small-sized, fast-inference-speed student model (which is non-autoregressive) by distilling knowledge from a larger, slower-inference-speed teacher model (which is autoregressive). A novel "Targeted Knowledge Distillation Framework" (TKDF) is introduced. This framework incorporates an evaluator and employs a curriculum learning approach to choose appropriate targets for the student model. Experiments reveal that the distilled student model, though smaller in size, matches the performance of the larger model while being 4.5 times faster in inference.

**Soundness Scores**: (3, 3, 2).
The paper's soundness is good. It includes results from two benchmarks. In the rebuttal phase, the authors introduced another benchmark result. TKDF exhibits a performance advantage over standard knowledge distillation, even though their inference speeds are comparable. However, to bolster the paper's claims, the following enhancements are suggested:
1. A comprehensive analysis of the model across all test benchmarks.
2. Inclusion of detailed experimental outcomes in the appendix.

**Excitement Scores**: (3, 3, 3).
From an excitement standpoint, the paper doesn't particularly stand out. The rationale for employing TKDF to tackle the SLU issue isn't convincingly articulated in the manuscript. As a result, the proposed TKDF appears to be a modest incremental improvement over existing methods.